# Minimax Bounds for Generalized Linear Models

**Kuan-Yun Lee and Thomas A. Courtade**
Department of Electrical Engineering and Computer Sciences
University of California, Berkeley
{timkylee,courtade}@berkeley.edu

## Abstract

We establish a new class of minimax prediction error bounds for generalized linear models. Our bounds significantly improve previous results when the design matrix is poorly structured, including natural cases where the matrix is wide or does not have full column rank. Apart from the typical $L_2$ risks, we study a class of entropic risks which recovers the usual $L_2$ prediction and estimation risks, and demonstrate that a tight analysis of Fisher information can uncover underlying structural dependency in terms of the spectrum of the design matrix. The minimax approach we take differs from the traditional metric entropy approach, and can be applied to many other settings.

## 1  Introduction

Throughout, we consider a parametric framework where observations $X \in \mathbb{R}^n$ are generated according to $X \sim P_\theta$, where $P_\theta$ denotes a probability measure on a measurable space $(\mathcal{X} \subseteq \mathbb{R}^n, \mathcal{F})$ indexed by an underlying parameter $\theta \in \Theta \subset \mathbb{R}^d$. For each $P_\theta$, we associate a density $f(\cdot; \theta)$ with respect to an underlying measure $\lambda$ on $(\mathcal{X}, \mathcal{F})$ according to

$$dP_\theta(x) = f(x; \theta)d\lambda(x).$$

This setup contains a vast array of fundamental applications in machine learning, engineering, neuroscience, finance, statistics and information theory [1–10]. As examples, mean estimation [1], covariance and precision matrix estimation [2], phase retrieval [3,4], group or membership testing [5], pairwise ranking [10], can all be modeled in terms of parametric statistics. The central question to address in all of these problems is essentially the same: *how accurately can we infer the parameter $\theta$ given the observation $X$?*

One of the most popular parameteric families is the *exponential family*, which captures a rich variety of parametric models such as binomial, Gaussian, Poisson, etc. Given a parameter $\eta \in \mathbb{R}$, a density $f(\cdot; \eta)$ is said to belong to the exponential family if it can be written as

$$f(x; \eta) = g(x) \exp\left(\frac{\eta x - \Phi(\eta)}{s(\sigma)}\right). \tag{1}$$

Here, the parameter $\eta$ is the *natural parameter*, $g : \mathcal{X} \subseteq \mathbb{R} \to [0, \infty)$ is the *base measure*, $\Phi : \mathbb{R} \to \mathbb{R}$ is the *cumulant function*, and $s(\sigma) > 0$ is a variance parameter. The density $f(\cdot; \eta)$ is understood to be on a probability space $(\mathcal{X} \subseteq \mathbb{R}, \mathcal{F})$ with respect to a dominating $\sigma$-finite measure $\lambda$.

In this work, we are interested in the following generalized linear model (GLM), where observation $X \in \mathbb{R}^n$ is generated according to an exponential family with natural parameter equal to a linear transformation of the underlying parameter $\theta$. In other words,

$$f(x; \theta) = \prod_{i=1}^{n} \left\{ g(x_i) \exp\left(\frac{x_i \langle m_i, \theta \rangle - \Phi(\langle m_i, \theta \rangle)}{s(\sigma)}\right) \right\}, \tag{2}$$

for a real parameter $\theta := (\theta_1, \theta_2, \ldots, \theta_d) \in \mathbb{R}^d$ and a fixed design matrix $M \in \mathbb{R}^{n \times d}$, with rows given by the vectors $\{m_i\}_{i=1}^n \subset \mathbb{R}^d$.

The above model assumes each $X_i$ is drawn from its own exponential family, with respective natural parameters $\langle m_i, \theta \rangle$, $i = 1, 2, \ldots, n$. Evidently, this captures the classical (Gaussian) linear model $X = M\theta + Z$, where $f(\cdot; \theta)$ is taken to be the usual Gaussian density, and also captures a much broader class of problems including phase retrieval, matrix recovery and logistic regression. See [11–13] for history and theory of the generalized linear model.

In order to evaluate the performance of an estimator $\hat{\theta}$ (i.e., a measurable function of $X$), it is common to define a loss function $\mathcal{L}(\cdot, \cdot) : \mathbb{R}^d \times \mathbb{R}^d \longmapsto \mathbb{R}$ and analyze the loss $\mathcal{L}(\theta, \hat{\theta})$. A typical figure of merit is the constrained minimax risk $R(M, \Theta)$, defined as

$$R(M, \Theta) := \inf_{\hat{\theta}} \sup_{\theta \in \Theta} \mathcal{L}(\theta, \hat{\theta}).$$

In words, the minimax risk characterizes the worst-case risk under the specified loss $\mathcal{L}(\cdot, \cdot)$ achieved by the best estimator, with a constraint that $\theta$ belongs to a specified parameter space $\Theta$.

Two choices of the loss function $\mathcal{L}(\cdot, \cdot)$ give rise to the usual variants of $L_2$ loss:

1. **Estimation loss**, where the loss function $\mathcal{L}(\cdot, \cdot)$ is defined as

$$\mathcal{L}_1(\theta, \hat{\theta}) = \mathbb{E}\|\theta - \hat{\theta}\|^2 \quad \text{for all } \theta, \hat{\theta} \in \mathbb{R}^d. \tag{3}$$

2. **Prediction loss**, where the loss function $\mathcal{L}(\cdot, \cdot)$ is defined as

$$\mathcal{L}_2(\theta, \hat{\theta}) = \frac{1}{n}\mathbb{E}\|M\theta - M\hat{\theta}\|^2 \quad \text{for all } \theta, \hat{\theta} \in \mathbb{R}^d. \tag{4}$$

In this work, we shall approach things from an information theoretic viewpoint. In particular, we will bound minimax risk under entropic loss (closely connected to logarithmic loss in the statistical learning and information literature, see, e.g., [14–16]), from which $L_2$ estimates will follow. To start, let us review some of the key definitions in information theory. Suppose the parameter $\theta \in \mathbb{R}^d$ follows a prior $\pi$, a probability measure on $\mathbb{R}^d$ having density $\psi$ with respect to Lebesgue measure. The differential entropy $h(\theta)$ corresponding to random variable $\theta$ is defined as

$$h(\theta) := -\int_{\mathbb{R}^d} \psi(u) \log \psi(u) du.$$

Here and throughout, we will take logarithms with respect to the natural base, and assume all entropies exist (i.e., their defining integrals exist in the Lebesgue sense). The mutual information $I(\theta; X)$ between parameter $\theta \sim \pi$ and observation $X \sim P_\theta$ is defined as

$$I(\theta; X) := \int_{\mathbb{R}^d} \int_{\mathcal{X}} f(x; \theta) \log \frac{f(x; \theta)}{\int_{\mathbb{R}^d} f(x; \theta') d\pi(\theta')} d\lambda(x) d\pi(\theta).$$

The conditional entropy is defined as $h(\theta|X) := h(\theta) - I(\theta; X)$. The entropy power of a random variable $U$ is defined as $\exp(2h(U))$, and for any two random variables $U$ and $V$ with well-defined conditional entropy, the conditional entropy power is defined similarly as $\exp(2h(U|V))$.

Lower bounds on conditional entropy power can be translated into lower bounds of other losses, via tools in rate distortion theory [17]. To illustrate this, let's consider the following two Bayes risks, with suprema taken over all priors $\pi$ on the parameter space $\Theta \subseteq \mathbb{R}^d$, and infima taken over all valid estimators $\hat{\theta}$ (i.e., measurable functions of $X$).

1. **Entropic estimation loss**, where the Bayes risk is defined as

$$R_e(M, \Theta) := \inf_{\hat{\theta}} \sup_{\pi} \sum_{i=1}^n \exp\left(2h(\theta_i|\hat{\theta}_i)\right). \tag{5}$$

2. **Entropic prediction loss**, where the Bayes risk is defined as

$$R_p(M, \Theta) := \inf_{\hat{\theta}} \sup_{\pi} \frac{1}{n} \sum_{i=1}^n \exp\left(2h(m_i^\top \theta | m_i^\top \hat{\theta})\right). \tag{6}$$

The following simple observation shows that any lower bound derived for the entropic Bayes risks implies a lower bound on the minimax $L_2$ risks.

**Lemma 1.** *We have* $\inf_{\hat{\theta}} \sup_{\theta \in \Theta} \mathcal{L}_1(\theta, \hat{\theta}) \gtrsim R_e(M, \Theta)$ *and* $\inf_{\hat{\theta}} \sup_{\theta \in \Theta} \mathcal{L}_2(\theta, \hat{\theta}) \gtrsim R_p(M, \Theta)$.

*Proof.* This follows since Gaussians maximize entropy subject to second moment constraints and conditioning reduces entropy: $\mathbb{E}(\theta_i - \hat{\theta}_i)^2 \geq \mathrm{Var}(\theta_i - \hat{\theta}_i) \gtrsim \exp(2h(\theta_i - \hat{\theta}_i)) \gtrsim \exp(2h(\theta_i | \hat{\theta}_i))$. □

Here and onwards, we use "$\gtrsim$" (also "$\lesssim$" and "$\asymp$") to refer to "$\geq$" (and "$\leq$", "$=$", respectively) up to constants that do not depend on parameters.

Although we focus on $L_2$ loss in the present work, we remark that minimax bounds on entropic loss directly yield corresponding estimates on $L_p$ loss using standard arguments involving covering and packing numbers of $L_p$ spaces. See, for example, the work by Raskutti et al. [18]. Despite its universal nature, there is relatively limited work on deriving minimax bounds for the entropic loss. This is the focus of the present work, and as a consequence, we obtain bounds on $L_2$ loss that significantly improve on prior results when the matrix $M$ is poorly structured.

## 1.1 Contributions

In this paper, we make three main contributions.

1. First, we establish $L_2$ minimax risk and entropic Bayes risk bounds for the generalized linear model (2). The generality of the GLM allows us to extend our results to specific instances of the GLM such as the Gaussian linear model, phase retrieval and matrix recovery.

2. Second, we establish $L_2$ minimax risk and entropic Bayes risk bounds for the Gaussian linear model. In particular, our bounds are nontrivial for many instances where previous results fail (for example when $M \in \mathbb{R}^{n \times d}$ does not have full column rank, including cases with $d > n$), and can be naturally applied to the sparse problem where $\|\theta\|_0 \leq k$. Further, we show that both our minimax risk and entropic Bayes risk bounds are tight up to constants and log factors when $M$ is sampled from a Gaussian ensemble.

3. Third, we investigate the $L_2$ minimax risk via the lens of the entropic Bayes risk, and provide evidence that information theoretic minimax methods can naturally extract dependencies on the structure of design matrix $M$ via analysis of Fisher information. The techniques we develop are general and can be used to establish minimax results for other problems.

## 2 Main Results and Discussion

The following notation is used throughout: upper-case letters (e.g., $X, Y$) denote random variables or matrices, and lower-case letters (e.g., $x, y$) denote realizations of random variables or vectors. We use subscript notation $v_i$ to denote the $i$-th component of a vector $v = (v_1, v_2, \ldots, v_d)$. We let $[k]$ denote the set $\{1, 2, \ldots, k\}$.

We will be making the following assumption.

**Assumption:** The second derivative of the cumulant function $\Phi$ is bounded uniformly by a constant $L > 0$: $\Phi''(\cdot) \leq L$.

The following lemma characterizes the mean and variance of densities in the exponential family.

**Lemma 2** (Page 29, [11]). *Any observation $X$ generated according to the exponential family* (1) *has mean $\Phi'(\eta)$ and variance $s(\sigma) \cdot \Phi''(\eta)$.*

In other words, our assumption is equivalent to saying that the variance of each observation $X_1, \ldots, X_n$ is bounded. This is a common assumption made in the literature; See, for example, [19–22].

Our first main result establishes a minimax prediction lower bound corresponding to the generalized linear model (2). Let us first make a few definitions. For an $n \times k$ matrix $A$, we define the vector $\Lambda_A := (\lambda_1, \ldots, \lambda_k) \in \mathbb{R}^k$, where the $\lambda_i$'s denote the eigenvalues of the $k \times k$ symmetric matrix

$A^\top A$ in descending order. $\|\Lambda_A\|_p$ denotes the usual $L_p$ norm of the vector $\Lambda_A$ for $p \geq 1$. Finally, we define

$$\Gamma(A) := \max\left( \frac{\|\Lambda_A\|_1^2}{\|\Lambda_A\|_2^2}, \, \lambda_{\min}(A^\top A) \|\Lambda_A^{-1}\|_1 \right), \tag{7}$$

where $\Lambda_A^{-1} := (\lambda_1^{-1}, \ldots, \lambda_k^{-1})$, with the convention that $\lambda_{\min}(A^\top A)\|\Lambda_A^{-1}\|_1 = 0$ when $\lambda_{\min}(A^\top A) = 0$.

**Theorem 3.** *For observations $X \in \mathbb{R}^n$ generated via the generalized linear model (2) with a fixed design matrix $M \in \mathbb{R}^{n \times d}$, the minimax $L_2$ prediction risk and the entropic Bayes prediction risk are lower bounded by*

$$\frac{1}{n} \inf_{\hat{\theta}} \sup_{\theta \in \mathbb{R}^d} \mathbb{E}\|M\hat{\theta} - M\theta\|^2 \gtrsim \frac{1}{n} \frac{s(\sigma)}{L} \Gamma(M).$$

$$\frac{1}{n} \inf_{\hat{\theta}} \sup_{\pi} \sum_{i=1}^n \exp\left( 2h(m_i^\top \theta \,|\, m_i^\top \hat{\theta}) \right) \gtrsim \frac{1}{n} \frac{s(\sigma)}{L} \frac{\|\Lambda_M\|_1^2}{\|\Lambda_M\|_2^2}.$$

Bounds on minimax risk under an additional sparsity constraint $\|\theta\|_0 \leq k$ (i.e., the true parameter $\theta$ has at most $k$ non-zero entries) can be derived as a corollary.

**Corollary 4** (Sparse Version of Theorem 3). *For observations $X \in \mathbb{R}^n$ generated via the generalized linear model (2), with the additional constraint that $\|\theta\|_0 \leq k$ (i.e., $\Theta := \{\theta \in \mathbb{R}^d : \|\theta\|_0 \leq k\}$), the minimax prediction error is lower bounded by*

$$\frac{1}{n} \inf_{\hat{\theta}} \sup_{\theta \in \Theta} \mathbb{E}\|M\hat{\theta} - M\theta\|^2 \gtrsim \frac{1}{n} \frac{s(\sigma)}{L} \max_{Q \in \mathcal{M}_k} \Gamma(Q).$$

$$\frac{1}{n} \inf_{\hat{\theta}} \sup_{\pi} \sum_{i=1}^n \exp\left( 2h(m_i^\top \theta \,|\, m_i^\top \hat{\theta}) \right) \gtrsim \frac{1}{n} \frac{s(\sigma)}{L} \max_{Q \in \mathcal{M}_k} \frac{\|\Lambda_Q\|_1^2}{\|\Lambda_Q\|_2^2}.$$

*Here, the maximum is taken over $\mathcal{M}_k$, the set of all $n \times k'$ submatrices of $M$, with $k' \leq k$.*

We now note an important specialization of Corollary 4. In particular, consider the Gaussian linear model with observations $X \in \mathbb{R}^n$ generated according to

$$X = M\theta + Z, \tag{8}$$

with $Z \sim \mathcal{N}(0, \sigma^2 \, \mathrm{I}_n)$ the standard Gaussian vector. This corresponds to the GLM of (2) when the functions are taken to be $h(x) = e^{-x^2/(2\sigma^2)}$, $s(\sigma) = \sigma^2$, and $\Phi(t) = t^2/2$ (hence, $L = 1$). This is a particularly important instance worth highlighting because of the ubiquity of the Gaussian linear model in applications.

**Theorem 5.** *For observations $X \in \mathbb{R}^n$ generated via the Gaussian linear model (8), with the sparsity constraint $\|\theta\|_0 \leq k$ (i.e., $\Theta := \{\theta \in \mathbb{R}^d : \|\theta\|_0 \leq k\}$), the minimax prediction error is lower bounded by*

$$\frac{1}{n} \inf_{\hat{\theta}} \sup_{\theta \in \Theta} \mathbb{E}\|M\hat{\theta} - M\theta\|^2 \gtrsim \frac{\sigma^2}{n} \max_{Q \in \mathcal{M}_k} \Gamma(Q).$$

$$\frac{1}{n} \inf_{\hat{\theta}} \sup_{\pi} \sum_{i=1}^n \exp\left( 2h(m_i^\top \theta \,|\, m_i^\top \hat{\theta}) \right) \gtrsim \frac{\sigma^2}{n} \max_{Q \in \mathcal{M}_k} \frac{\|\Lambda_Q\|_1^2}{\|\Lambda_Q\|_2^2}.$$

*Here, the maximum is taken over $\mathcal{M}_k$, the set of all $n \times k'$ submatrices of $M$, with $k' \leq k$.*

**Remark 6.** *In the above results, the function $\Gamma(\cdot)$ can in fact be replaced with*

$$\tilde{\Gamma}(M) := \max\left( \sum_{i=1}^n \frac{\|m_i\|_2^4}{\|Mm_i\|^2}, \, \lambda_{\min}(M^\top M)\|\Lambda_M^{-1}\|_1 \right),$$

*which is stronger than the original statements. However, the chosen statements above highlight the simple dependence on the spectrum of $\Lambda_M$.*

## 2.1 Related Work

Most relevant to our results is the following lower bound on minimax $L_2$ estimation risk and entropic Bayes estimation risk, developed in a recent work by Lee and Courtade [23]. We note that [23] does not bound prediction loss (which is often of primary interest), as we have done in the present paper.

**Theorem 7** (Theorem 3, [23]). *Let observation $X$ be generated via the generalized linear model defined in* (2), *with the additional structural constraint $\Theta = \mathbb{B}_2^d(R) := \{v : \|v\|_2^2 \leq R^2\}$. Suppose the cumulant function $\Phi$ satisfies $\Phi'' \leq L$ for some constant L. Then, the minimax estimation error is lower bounded by*

$$\inf_{\hat{\theta}} \sup_{\theta \in \Theta} \mathbb{E}\|\hat{\theta} - \theta\|^2 \gtrsim \inf_{\hat{\theta}} \sup_{\pi} \sum_{i=1}^n \exp(2h(\theta_i|\hat{\theta}_i)) \gtrsim \min\left(R^2, \frac{s(\sigma)}{L} \operatorname{Tr}((M^\top M)^{-1})\right). \quad (9)$$

The bound of (9) is tight when $X$ is generated by the Gaussian linear model, showing that (Gaussian) linear models are most favorable in the sense of minimax estimation error amongst the class of GLMs considered here. Lee and Courtade extracted the dependence on the $\operatorname{Tr}(M^\top M)$ term by analyzing a Fisher information term in the class of Bayesian Cramér-Rao-type bounds from [24]. Earlier work (see, e.g., [25]) yielded bounds on the order of $d/\lambda_{\max}(M^\top M)$, which is loose compared to (9).

There is a large body of work that establish minimax lower bounds on prediction error for specific models of the generalized linear model. Typically, these analyses depend on methods involving metric entropy (see, for example, [4, 18, 19, 26–28]). A popular minimax result is due to Raskutti et al. [18], who consider the sparse Gaussian linear model, where for a fixed design matrix $M$ with an additional sparsity constraint $\|\theta\|_0 \leq k$,

$$\sigma^2 \frac{\Phi_{2k,-}(M)}{\Phi_{2k,+}(M)} \frac{k}{n} \log\left(\frac{ed}{k}\right) \lesssim \inf_{\hat{\theta}} \sup_{\|\theta\|_0 \leq k} \frac{1}{n} \mathbb{E}\|M\hat{\theta} - M\theta\|_2^2 \lesssim \sigma^2 \min\left(\frac{k}{n} \log\left(\frac{ed}{k}\right), 1\right). \quad (10)$$

Here the terms $\Phi_{r,-}(M)$ and $\Phi_{r,+}(M)$ correspond to the *constrained eigenvalues*,

$$\Phi_{r,-}(M) := \inf_{0 \neq \|\theta\|_0 \leq r} \frac{\|M\theta\|^2}{\|\theta\|^2}, \qquad \Phi_{r,+}(M) := \sup_{0 \neq \|\theta\|_0 \leq r} \frac{\|M\theta\|^2}{\|\theta\|^2}. \quad (11)$$

The upper bound of (10) is achieved by classical methods such as aggregation [29–32].

One can readily observe that the lower bound of (10) becomes degenerate for even mildly ill-structured design matrices $M$. For example, in the case where $M$ has repeating columns, the above result gives a lower bound of $0$, which is not very interesting. This suggests that the metric entropy approach does not easily capture the dependence of the structure of design matrix $M$ at the resolution of the complete spectrum of $M^\top M$ as our results do. In fact, it can be shown that Corollary 4 uniformly improves upon (10) up to logarithmic factors; see Section 4.1 of the supplementary. Further, the lower bound of Raskutti et al. does not hold for $k > n$, which is a disadvantage for high dimensional problems where $d \gg n$. Verzelen [30] discusses the regime where $\frac{k}{n} \log\left(\frac{ed}{k}\right) \geq \frac{1}{2}$ and $k \leq \max(d^{1/3}, n/5)$ and provide bounds for the worst-case matrix $M$, which is a different setting from ours.

There are also lines of work on specific settings of the generalized linear model. For example, Candes et al. [28] discusses low-rank matrix recovery, and Cai et al. [4] considers phase retrieval. There are, however, fewer results that directly look at the generalized linear model of our setting. The closest work related is that of Abramovich and Grinshtein [19], where they consider estimating the entire vector $M\theta$, as opposed to our setting where we estimate $\theta$ first with $\hat{\theta}$, then evaluate $M\hat{\theta}$. Their result also depends on the ratio between (constrained) minimum and maximum eigenvalues as in (10), and hence fails when $M$ is not full rank or otherwise has divergent maximum and minimum (constrained) eigenvalues.

Comparing Theorems 3 and 5 with the results surveyed above raises several points (illustrated in Table 1):

- **Nontrivialness when $M$ is not full rank.** Unlike the lower bound in (10), the ratio $\|\Lambda_M\|_1^2/\|\Lambda_M\|_2^2$ does not vanish when $M$ is not full rank; see Case (d) in Table 1. This is particularly important when the dimension of the parameter is large relative to the number of observed samples.

Table 1: Values of identities in $\Gamma(M)$ (defined in (7)) for different scenarios of $\Lambda_M = (\lambda_1, \ldots, \lambda_d)$ for fixed $M \in \mathbb{R}^{n \times d}$. The value $t$ satisfies $t \gg 1$. In each row, the bold item marks the largest value.

| Case | $\|\Lambda_M\|_1^2/\|\Lambda_M\|_2^2$ | $\lambda_d\|\Lambda_M^{-1}\|_1$ | $d(\lambda_d/\lambda_1)$ |
|---|---|---|---|
| (a) $\Lambda_M = (1, 1, \ldots, 1, 1)$ | $\boldsymbol{d}$ | $\boldsymbol{d}$ | $\boldsymbol{d}$ |
| (b) $\Lambda_M = (t, 1, 1, \ldots, 1)$ | $\approx (t+d)^2/(t^2+d)$ | $\boldsymbol{\approx d}$ | $d/t$ |
| (c) $\Lambda_M = (1, 1, \ldots, 1, 1/t)$ | $\boldsymbol{\approx d}$ | $\approx d/t$ | $d/t$ |
| (d) $\Lambda_M = (1, 1, \ldots, 1, 0)$ | $\boldsymbol{\approx d}$ | $0$ | $0$ |
| (e) $\Lambda_M = (t, 1, \ldots, 1, 1/t)$ | $\boldsymbol{\approx (t+d)^2/(t^2+d)}$ | $\approx (t+d)/t$ | $d/t^2$ |

- **Insensitivity to extreme values in the spectrum $\Lambda_M$.** Unlike the ratio between largest and smallest (restricted) eigenvalues in $\Lambda_M$, the ratio $\|\Lambda_M\|_1^2/\|\Lambda_M\|_2^2$ is less sensitive to the setting where the maximum and minimum eigenvalues in $\Lambda_M$ diverge. Table 1 provides several examples in rows (b)-(e).

- **Sharpness.** Comparing with the upper bound (10), we observe that Theorem 3 (or, more specifically, Theorem 5) is sharp if either the largest (constant·$d$) eigenvalues are of the same order or (for the $L_2$ risk) if the smallest (constant·$d$) eigenvalues are non-zero and of the same order. This can be seen by considering $\|\Lambda_M\|_1^2/\|\Lambda_M\|_2^2$ and $\lambda_{\min}(M^\top M)\|\Lambda_M^{-1}\|_1$ in the former and latter cases, respectively. Moreover, as shown in the following Section, when $M$ is sampled from a Gaussian ensemble, i.e., all components of $M$ are sampled from a standard Gaussian, our bounds are optimal up to log factors with high probability.

- **Logarithmic term for sparse linear regression.** In many cases, the log factor is insignificant, and the improved spectral dependence of Theorem 5 can yield substantial improvement. For example, when $k$ is not very sparse, say $k = \Theta(d^c)$ for some $c \in (0, 1]$, the log factor is not significant and our results can be significantly better than (10) when $M$ is mildly ill-conditioned. In the very sparse case, say $k = O(\log d)$, our results still provide meaningful bounds for $M$ with minimum constrained eigenvalue close to 0.

**Remark 8.** *In some cases, (10) can be improved by ignoring certain components of $\theta \in \mathbb{R}^d$ via dimensionality reduction. For example, if the first two columns of $M$ are the same, then it is possible to ignore the first component of $\theta$ and simply look at the remaining $d-1$ components. We remark that even with this reduction, (10) still depends on the ratio between minimum and maximum constrained eigenvalues of the new "effective" matrix, and leads to a poor lower bound when the minimum and maximum constrained eigenvalues are of a different order. We remark that other dimensionality reduction methods (such as rotations) may be limited by the sparsity constraint $\|\theta\|_0 \le k$. Moreover, in general when the spectrum of $M$ is all positive (with divergent large/small eigenvalues), one cannot use dimensionality reduction to improve the result of (10).*

## 2.2 Application to Gaussian Designs

Gaussian designs are frequently adopted in machine learning and compressed sensing (see, for example, [18, 33–35]). The following proposition provides a concentration bound for the ratio $\|\Lambda_M\|_1^2/\|\Lambda_M\|_2^2$ when $M$ is sampled from the standard Gaussian ensemble (i.e., where each component of $M$ is sampled i.i.d. according to a standard Gaussian).

**Proposition 9.** *Let the design matrix $M \in \mathbb{R}^{n \times k}$ be sampled from the Gaussian ensemble. There exist universal constants $c_1, c_2, c_3 > 0$ such that $\|\Lambda_M\|_1^2/\|\Lambda_M\|_2^2 \ge c_1 \min(n, k)$ with probability at least $1 - c_2 \exp(-c_3 \min(n, k))$.*

Proposition 9 implies that, with high probability, the lower bound of Theorem 5 (and therefore the corresponding estimate in Theorem 3) is sharp up to a logarithmic term that is negligible when $d \asymp k$. In particular, under the assumptions of Theorem 5, we obtain with the help of (10) that

$$\sigma^2 \min\left(\frac{k}{n}, 1\right) \lesssim \inf_{\hat\theta} \sup_{\|\theta\|_0 \le k} \frac{1}{n} \mathbb{E}\|M\hat\theta - M\theta\|_2^2 \lesssim \sigma^2 \min\left(\frac{k}{n}\log\left(\frac{ed}{k}\right), 1\right), \qquad (12)$$

with the lower bound holding with high probability in $\min(n, k)$. This can significantly improve on the lower bound (10); consider, for example, the case where $s := \min(2k, d) = \alpha n$ for

some fixed $\alpha < 1$. Note that any $n \times s$ submatrix $M'$ of $M$ satisfies $\Phi_{2k,-}(M)/\Phi_{2k,+}(M) \leq \lambda_{\min}(M'^{\top}M')/\lambda_{\max}(M'^{\top}M')$. An asymptotic result by Bai and Yin [36] implies that if $\alpha$ is fixed then this latter ratio converges to $(1 - \sqrt{\alpha})^2 / (1 + \sqrt{\alpha})^2$ almost surely as $n, k, d \to \infty$. Hence, asymptotically speaking, the result of (10) is tight at most up to constants depending on $\alpha$ while our results of Corollary 4 is tight (up to log factors) without dependency of $\alpha$.

Interestingly, Proposition 9 also holds for square matrices, where the minimum eigenvalue is close to zero (more precisely, for a square Gaussian matrix $M \in \mathbb{R}^{n \times n}$, $\lambda_{\min}(M^{\top}M)$ is of the order $n^{-1}$, as shown in the work of Rudelson and Vershynin [37]). Proposition 9 follows from Szarek's work [38] on concentration of the largest $n/2$ singular values for a square Gaussian matrix $M \in \mathbb{R}^{n \times n}$, concentration of singular values of rectangular subgaussian matrices [26], and an application of interlacing inequalities for singular values of submatrices [39]. Similar results can be shown for subgaussian matrices under additional assumptions using tools from [40].

## 3   Key Points of Proofs of Main Theorems

In our approach, we will be using classical information theory tools inspired by the techniques developed by Lee and Courtade [23].

### 3.1   Preliminaries

We say that a measure $\mu$ is log-concave if $d\mu(x) = e^{-V(x)}dx$ for some convex function $V(\cdot)$. The Fisher information $\mathcal{I}_X(\theta)$ given $\theta \in \mathbb{R}^d$ corresponding to the map $\theta \longmapsto P_\theta$ is defined as

$$\mathcal{I}_X(\theta) = \mathbb{E}_X \|\nabla_\theta \log f(X;\theta)\|_2^2 ,$$

where the gradient is taken with respect to $\theta$, and the expectation is taken with respect to $X \sim P_\theta$. If the parameter $\theta$ has a prior $\pi$ that is log-concave, the following lemma gives an upper bound on the mutual information $I(\theta;X)$, which depends on the covariance matrix of $\theta$, defined as $\mathrm{Cov}(\theta)$.

**Lemma 10** (Theorem 2, [24]). *Suppose the prior $\pi$ of $\theta \in \mathbb{R}^d$ is log-concave. Then, under mild regularity conditions on the map $\theta \longmapsto P_\theta$, we have*

$$I(\theta;X) \leq d \cdot \phi\left(\frac{\mathrm{Tr}(\mathrm{Cov}(\theta)) \cdot \mathbb{E}\,\mathcal{I}_X(\theta)}{d^2}\right), \tag{13}$$

*where the function $\phi(\cdot)$ is defined as $\phi(x) := \begin{cases} \sqrt{x} & \text{if } 0 \leq x < 1, \\ 1 + \frac{1}{2}\log x & \text{if } x \geq 1. \end{cases}$*

We note that the regularity condition in Lemma 10 requires that each member of the parametric family $P_\theta$ has density $f(\cdot;\theta)$ smooth enough to permit the following change of integral and differentiation,

$$\int_{\mathcal{X}} \nabla_\theta f(x;\theta)d\lambda(x) = 0, \quad \mu - a.e.\ \theta. \tag{14}$$

In our case, since we are working with the GLM of (2), the regularity condition is automatically satisfied.

When $\theta$ is a one-dimensional (i.e., $d = 1$) log-concave random variable, the bound of (13) is sharp up to a (modest) multiplicative constant when $\mathrm{Var}(\theta)\,\mathbb{E}\,\mathcal{I}_X(\theta)$ is bounded away from zero. There exists a tighter version of Lemma 10 when $\pi$ is uniformly log-concave, however Lemma 10 is enough for our purposes. We direct the interested reader to the paper [24].

### 3.2   Proof Sketch of Theorem 3

We start off by noting that we can lower bound the entropic Bayes risk of (6) by taking a specific prior $\pi$. For our purposes, we will let $\theta$ have a multivariate Gaussian prior $\pi = \mathcal{N}\left(0, \beta^2 \mathrm{I}_d\right)$.

We continue with a bound on the sum of conditional entropy powers

$$\sum_{i=1}^{n} \exp\left(2h(m_i^{\top}\theta \mid m_i^{\top}\hat{\theta})\right) \geq \sum_{i=1}^{n} \exp\left(2h(m_i^{\top}\theta) - 2I(m_i^{\top}\theta;X)\right), \tag{15}$$

which follows from the data-processing inequality $I(m_i^\top \theta; m_i^\top \hat{\theta}) \leq I(m_i^\top \theta; X)$, since $m_i^\top \theta \rightarrow X \rightarrow m_i^\top \hat{\theta}$ forms a Markov chain.

When $m_i \in \mathbb{R}^d$ is a zero-vector, $\exp\left(2h(m_i^\top \theta | m_i^\top \theta)\right) = \exp\left(2h(m_i^\top \theta) - 2I(m_i^\top \theta; X)\right) = 0$ and hence does not contribute to the summations within (15). This implies that removing zero vector rows from $M$ does not affect the proof following (15). Hence, in the following we will assume that the matrix $M$ does not have rows that are zero vectors.

By our choice of the prior $\pi$, the density of $m_i^\top \theta$ is Gaussian and hence log-concave, which allows us to invoke Lemma 10, implying

$$\sum_{i=1}^{n} \exp\left(2h(m_i^\top \theta \,|\, m_i^\top \hat{\theta})\right) \geq \sum_{i=1}^{n} \exp\left(2h(m_i^\top \theta) - 2\phi(\mathrm{Var}(m_i^\top \theta) \cdot \mathbb{E}\,\mathcal{I}_X(m_i^\top \theta))\right). \quad (16)$$

Here, the expectation is taken with respect to the marginal density of $m_i^\top \theta$. The primary task is now to obtain a reasonable bound on the expected Fisher information term $\mathbb{E}\,\mathcal{I}_X(m_i^\top \theta)$. To do this, we introduce the following lemma, which provides an upper bound for the expected Fisher information $\mathbb{E}\,\mathcal{I}_X(m_i^\top \theta)$.

**Lemma 11.** *Fix $M \in \mathbb{R}^{n \times d}$. If parameter $\theta$ has a prior $\pi = \mathcal{N}(0, \beta^2 I_d)$ and $X \in \mathbb{R}^n$ is sampled according to the generalized linear model defined as* (2)*, then*

$$\mathbb{E}\,\mathcal{I}_X(m_i^\top \theta) \leq \frac{L}{s(\sigma)} \cdot \frac{\|Mm_i\|_2^2}{\|m_i\|_2^4} + \frac{1}{\beta^2} \cdot \Psi_i(M) \qquad \textit{for all } i = 1, 2, \ldots, n. \quad (17)$$

*The function $\Psi_i(M)$ depends only on $M$ and is finite. The expectation is taken with respect to the marginal density of $m_i^\top \theta$.*

The functions $\Psi_i(\cdot)$ are not explictly stated here because later we will be taking $\beta$ large enough so that $\Psi_i(\cdot)/\beta^2$ in (17) can be ignored. A proof of Lemma 11 and more details about the functions $\Psi_i(\cdot)$ are included in the supplementary. We can continue from (16) and see that

$$\sum_{i=1}^{n} \exp\left(2h(m_i^\top \theta \,|\, m_i^\top \hat{\theta})\right)$$
$$\gtrsim \sum_{i=1}^{n} \beta^2 \|m_i\|_2^2 \exp\left(-2\phi\left[\beta^2 \|m_i\|_2^2 \left(\frac{L}{s(\sigma)} \cdot \frac{\|Mm_i\|_2^2}{\|m_i\|_2^4} + \frac{1}{\beta^2}\Psi_i(M)\right)\right]\right)$$
$$\overset{(a)}{\gtrsim} \sum_{i=1}^{n} \frac{1}{\frac{L}{s(\sigma)} \cdot \frac{\|Mm_i\|_2^2}{\|m_i\|_2^4} + \frac{1}{\beta^2}\Psi_i(M)} \overset{(b)}{=} (1-\epsilon) \frac{s(\sigma)}{L} \sum_{i=1}^{n} \frac{\|m_i\|_2^4}{\|Mm_i\|_2^2}. \quad (18)$$

In the above, both (a) and (b) require a selection of $\beta^2$ to be large enough. In particular, in (a), $\beta^2 \geq s(\sigma)/L$ would guarantee that the function $\phi$ behaves logarithmically (recall from Lemma 10 that $\phi(t)$ behaves logarithmically if $t \geq 1$). In (b), the variable $\epsilon$ depends on the selection of $\beta$. Since the function $\Psi_i(M)$ is finite for all $i = 1, \ldots, n$, by taking $\beta^2$ a constant large enough, we can force $\epsilon$ to be as close to zero as possible. Hence, we can say that the inequality holds with $\epsilon = 0$. A direct application of the Cauchy-Schwarz inequality then yields

$$\sum_{i=1}^{n} \exp\left(2h(m_i^\top \theta \,|\, m_i^\top \hat{\theta})\right) \geq \frac{s(\sigma)}{L} \frac{\left(\sum_{i=1}^{n} \|m_i\|_2^2\right)^2}{\sum_{i=1}^{n} \|Mm_i\|_2^2} = \frac{s(\sigma)}{L} \frac{\|\Lambda_M\|_1^2}{\|\Lambda_M\|_2^2}. \quad (19)$$

On the other hand, from Theorem 7 and the matrix identity $\|Mv\|_2^2 \geq \lambda_{\min}(M^\top M)\|v\|_2^2$,

$$\inf_{\hat{\theta}} \sup_{\theta \in \mathbb{R}^d} \mathbb{E}\|M\hat{\theta} - M\theta\|_2^2 \geq \lambda_{\min}(M^\top M) \cdot \mathrm{Tr}\left((M^\top M)^{-1}\right) = \lambda_d \|\Lambda_M^{-1}\|_1. \quad (20)$$

Combining (19) and (20) with Lemma 1 finishes the proof.

### 3.3 An Alternative Proof of Theorem 5

For the Gaussian linear model, we have the following tighter version of Lemma 11.

**Lemma 12.** *Fix $M \in \mathbb{R}^{n \times d}$. If $\theta \sim \mathcal{N}(0, \beta^2 I_d)$ and $X \in \mathbb{R}^n$ is sampled according to the Gaussian linear model defined as (8). Then,*

$$\mathbb{E}\,\mathcal{I}_X(m_i^\top \theta) \leq \frac{1}{\sigma^2} \cdot \frac{\|Mm_i\|_2^2}{\|m_i\|_2^4} \qquad for \ 1 \leq i \leq n. \tag{21}$$

By taking any $\beta^2 \geq \sigma^2 \max_i \left( \|m_i\|_2^2 / \|Mm_i\|_2^2 \right)$, the function $\phi(\cdot)$ in (16) will again behave logarithmically, directly implying (18) with $\epsilon = 0$. The remaining proof follows similarly as before.

**Remark 13.** *The functions $\Psi_i(\cdot)$ can be difficult to bound directly (see supplementary for more details). Hence, the improved tightness and simplicity of Lemma 12 over Lemma 11 for the Gaussian linear model provides more flexibility on the selection of $\beta$. This can be helpful when dealing with problem settings where there are other constraints on the parameter space $\Theta$.*

**Remark 14.** *There is a subtle but crucial difference in the proof techniques employed here compared to those in [23]. The key step in [23] requires bounding the Fisher information $\mathcal{I}_X(\theta_i)$ with diagonal terms in the Fisher information matrix $\mathcal{I}_X(\theta)$, i.e., Lemma 9 of [23]. In our case, we need to bound the Fisher information $\mathcal{I}_X(m_i^\top \theta)$ (e.g., Lemma 11), and here, the terms $m_i^\top \theta$ are not necessarily mutually independent as required in Lemma 9 of [23], which prevents us from a direct application. Instead, we choose $\theta$ to have a Gaussian prior and try to bound $\mathcal{I}_X(\theta_i)$ directly. This is facilitated by properties of the Gaussian distribution; see Section 4.3 in the appendix for more details.*

## Broader Impact

The generalized linear model (GLM) is a broad class of statistical models that have extensive applications in machine learning, electrical engineering, finance, biology, and many areas not stated here. Many algorithms have been proposed for inference, prediction and classification tasks under the umbrella of the GLM, such as the Lasso algorithm, the EM algorithm, Dantzig selectors, etc., but often it is hard to confidently assess optimality. Lower bounds for minimax and Bayes risks play a key role here by providing theoretical benchmarks with which one can evaluate the performance of algorithms. While many previous approaches have focused on the Gaussian linear model, in this paper we establish minimax and Bayes risk lower bounds that hold uniformly over all statistical models within the GLM. Our arguments demonstrate a set of information-theoretic techniques that are general and applicable to setups other than the GLM. As a result, many applications stand to potentially benefit from our work.

## Acknowledgments

This work was supported in part by NSF grants CCF-1704967, CCF-1750430, CCF-0939370.

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
