[Supplementary Material]

# 4 Appendix

In the appendix, we provide additional proofs that have been omitted from the main text.

## 4.1 Uniform improvement of Corollary 4 over (10)

To see this, let $M'$ be the realization of $\Phi_{2k,+}(M)$ with most columns ($M'$ will have at least $k$ columns since adding a column to $M'$ will not decrease its maximum singular value). Note that $k\Phi_{2k,-}(M)/\Phi_{2k,+}(M) \leq k\sigma^2_{\min}(M')/\sigma^2_{\max}(M')$ by definition of $\Phi$. We may assume that $M'$ has full column rank, otherwise the proof is trivial. Then, for any submatrix $Q \in \mathbb{R}^{n\times k}$ of $M'$, we have $k\sigma^2_{\min}(M')/\sigma^2_{\max}(M') \leq k\sigma^2_{\min}(Q)/\sigma^2_{\max}(Q)$ (see, e.g., 2.2.33. of [41]). Next, $k\sigma^2_{\min}(Q)/\sigma^2_{\max}(Q) \leq \|\Lambda_Q\|^2_1/\|\Lambda_Q\|^2_2$ holds because $\sigma^2_{\max}(Q)\|\Lambda_Q\|^2_1 \geq k\sigma^2_{\min}(Q)(\sigma^2_{\max}(Q)\|\Lambda_Q\|_1) \geq k\sigma^2_{\min}\|\Lambda_Q\|^2_2$. Combining these with Corollary 4 gives the claim.

## 4.2 Proof of Proposition 9

We first focus on the case where $M \in \mathbb{R}^{n\times n}$ is a square matrix. Recall that we defined $\lambda_1 \geq \ldots \geq \lambda_n$ as the nonincreasing order of eigenvalues of matrix $M^\top M$.

The following result is due to Szarek [38].

**Lemma 15** (Theorem 1.3, [38])**.** *There exist universal constants $c_1 c_2, c', C'$ such that for any $k \leq (1/2)n$, with probability at least $1 - C' \exp\left(-c'k^2\right)$,*

$$c_1 k^2 \leq \lambda_{n-k}(M^\top M) \leq c_2 k^2.$$

We can take $k = n/2$ (or $(n-1)/2$ if $n$ is odd; in the following we will be assuming $n$ is even and the odd case follows similarly). It follows by the union bound that there are universal constants $c_3. c_4, c, C$ such that with probability at least $1 - C\exp(-cn^2)$,

$$n^2 c_3 \leq \lambda_{n/2}(M^\top M) \leq \lambda_1(M^\top M) \leq n^2 c_4. \tag{22}$$

Finally, we require the following lemma.

**Lemma 16.** *Suppose $A \geq X > 0$ and $B \geq Y > 0$. Then,*

$$\frac{(A+X)^2}{B+Y} \geq \frac{A^2}{2B}.$$

*Proof.* It suffices to show that $2A^2 B + 4ABX + 2BX^2 \geq A^2 B + A^2 Y$ which follows directly by $A^2 B \geq A^2 Y$ and non-negativity of $A, B, X$. $\qquad\square$

With Lemma 16 and (22), we see that with probability at least $1 - C\exp(-cn^2)$,

$$\frac{\|\Lambda\|^2_1}{\|\Lambda\|^2_2} \geq \frac{\left(\sum_{i=1}^{n/2} \lambda_i\right)^2}{\sum_{i=1}^{n/2} \lambda_i^2} \geq c_6 n.$$

where $c_6$ is a constant that depends on the universal constants $c_1, c_5$. We remark that it is easy to show that $c_6 = c_3^2/(2c_4^2)$ suffices, because

$$\left(\sum_{i=1}^{n/2} \lambda_i\right)^2 \geq \frac{n^6}{4}c_3^2 = \frac{c_3^2}{c_4^2} \cdot \frac{n^6}{4}c_4^2 \geq \frac{c_3^2}{2c_4^2} \cdot n \sum_{i=1}^{n/2} \lambda_i^2.$$

This completes the proof of Proposition 9 for the case where $M$ is a square matrix.

For a non-square matrix $M \in \mathbb{R}^{n\times d}$, we can without loss of generality assume that $d < n$, and the argument for $n > d$ follows similarly. Again, recall that we define $\lambda_1 \geq \ldots \lambda_d$ as the nonincreasing order of eigenvalues of the matrix $M^\top M$. Now, if $d \geq \frac{3}{4}n$, we invoke the following lemma which is a direct consequence of Theorem 1 of [39].

**Lemma 17.** *For any $d \times d$ submatrix $D$ of $M$, let $\beta_1 \geq \ldots \geq \beta_d$ be the eigenvalues of $D^\top D$. Then,*

$$\begin{cases} \lambda_i \geq \beta_i & \text{for } i = 1, 2, \ldots, d \\ \beta_i \geq \lambda_{i+n-d} & \text{for } i \leq 2d - n \end{cases}.$$

Lemma 17 and Lemma 15 imply that there exists constants $c_6, c_6', c_7, c_8, c_9, c'', C''$ such that $\lambda_{d/2} \geq \beta_{d/2} \geq c_6 d^2 = c_6' n^2$ with probability at least $1 - C' \exp\left(-c' d^2\right)$. On the other hand, we have $\lambda_1 \leq c_7 n^2$ with probability at least $1 - c_8 \exp\left(-c_9 n\right)$ from the following classical tail bound (see, e.g., [26])

$$P((\sqrt{n} - \sqrt{d} - t)^4 \leq \lambda_d(M^\top M) \leq \lambda_1(M^\top M) \leq (\sqrt{n} + \sqrt{d} - t)^4) \leq 2e^{-t^2/2}, \quad t \geq 0. \tag{23}$$

It follows that we can use a similar argument as in the square matrix setting to complete the proof. Finally, if $d < \frac{3}{4}n$, then our claim follows naturally by a direct analysis using (23).

## 4.3 Proof of Lemma 11

For any matrix $M \in \mathbb{R}^{n \times d}$, we can find an orthogonal matrix $Q \in \mathbb{R}^{d \times d}$ such that the first row of $MQ$ has only one non-zero component, and is in the first component. Then, the distribution of $M\theta$ is the same as the distribution as $MQQ^\top \theta$. Analyzing the Fisher information $\mathcal{I}_X(m_i^\top \theta)$ is equivalent to analyzing the Fisher information $\mathcal{I}_X(\tilde{m}_i^\top \tilde{\theta})$, where we take the transform $\tilde{M} = MQ$ and let $\tilde{m}_i$ be the $i$-th row of $\tilde{M}$. The random variable $\tilde{\theta} := Q^\top \theta$ follows a Gaussian distribution with covariance $\beta^2 Q^\top Q = \beta^2 \, \mathrm{I}_d$. Note that the norm of the first row $\tilde{m}_1$ is $\|\tilde{m}_1\|_2^2 = \|m_1^\top Q Q^\top m_1\|_2^2 = \|m_1\|_2^2$.

This implies that we can discuss with the matrix $\tilde{M}$ as our design matrix. We will do so, and drop the tilde sign for simplicity. In other words, we assume that $m_1$ is such that only the first component is non-zero.

Let $A = [m_2 \quad \ldots \quad m_n]^\top$. Then, the conditional distribution of $A\theta$ given $m_1^\top \theta = t$ is a multivariate Gaussian with mean and variance given by

$$\text{mean} = \underbrace{\frac{Am_1}{\|m_1\|_2^2}}_{:=\mu} \cdot t \qquad \Sigma = \left( AA^\top - \frac{Am_1 m_1^\top A^\top}{\|m_1\|_2^2} \right)\beta^2, \tag{24}$$

Here, we defined the vector $\mu$ as $\mu := Am_1/\|m_1\|_2^2$. We further define the following quantities: we define the function $q(s|t)$ as the conditional density of $A\theta = s$, $s \in \mathbb{R}^{n-1}$ given $m_1^\top \theta = t$, $t \in \mathbb{R}$, and define $f(x|\cdot)$ as the conditional density of $X$ given $(\cdot)$. We define $f'(\cdot|\cdot, m_1^\top \theta = t)$ and $q'(\cdot|m_1^\top \theta = t)$ as the corresponding derivatives with respect to $t$. Consequently, we may calculate the Fisher information $\mathcal{I}_X(m_i^\top \theta = t)$ as

$$\mathcal{I}_X(m_i^\top \theta = t)$$

$$= \int_\mathcal{X} \frac{[f'(x|m_1^\top \theta = t)]^2}{f(x|m_1^\top \theta = t)} d\lambda(x)$$

$$\overset{(a)}{=} \int_\mathcal{X} \frac{\left[\int_{\mathbb{R}^{n-1}} \left( f'(x|A\theta = s, m_1^\top \theta = t)q(s|t) + f(x|A\theta = s, m_1^\top \theta = t)q'(s|t) \right) ds\right]^2}{\int_{\mathbb{R}^{n-1}} f(x|A\theta = s, m_1^\top \theta = t)q(s|t)ds} d\lambda(x)$$

$$\overset{(b)}{\leq} \int_\mathcal{X} \int_{\mathbb{R}^{n-1}} \frac{\left[f'(x|A\theta = s, m_1^\top \theta = t)q(s|t) + f(x|A\theta = s, m_1^\top \theta = t)q'(s|t)\right]^2}{f(x|A\theta = s, m_1^\top \theta = t)q(s|t)} ds \, d\lambda(x)$$

$$\overset{(c)}{=} \int_\mathcal{X} \int_{\mathbb{R}^{n-1}} f(x|A\theta = s, m_1^\top \theta = t)q(s|t) \, \mathbf{T} \, ds \, d\lambda(x)$$

$$\overset{(d)}{=} \frac{L}{s(\sigma)} \cdot \sum_{i=1}^n \frac{(m_i^\top m_1)^2}{\|m_1\|_2^4} + \mu^\top \Sigma^\dagger \mu$$

$$= \frac{L}{s(\sigma)} \cdot \frac{\|Mm_1\|_2^2}{\|m_1\|_2^4} + \mu^\top \Sigma^\dagger \mu. \tag{25}$$

Here, we define $\mathbf{T}$ as

$$\mathbf{T} = \left( \frac{x_1 - \Phi'(t)}{s(\sigma)} \cdot \frac{m_1^\top m_1}{\|m_1\|_2^2} + \sum_{i=2}^n \left( \frac{x_i - \Phi'(s_{i-1})}{s(\sigma)} \cdot \frac{m_i^\top m_1}{\|m_1\|_2^2} \right) + \mu^\top \Sigma^\dagger (s - \mu t) \right)^2.$$

The term $\Sigma^\dagger$ corresponds to the pseudo-inverse of the covariance matrix $\Sigma$. Equality (a) is an exchange of integration and differentiation, justified by the Leibniz rule and smoothness of both the normal distribution and the regularity of the generalized linear model. Inequality (b) follows from Cauchy-Schwarz. Equality (c) follows from the observation that $f'(x|A\theta = s, m_1^\top \theta = t)$ can be written as

$$f'(x|A\theta = s, m_1^\top \theta = t)$$
$$= f(x|A\theta = s, m_1^\top \theta = t) \left( \frac{x_1 - \Phi'(t)}{s(\sigma)} \cdot \frac{m_1^\top m_1}{\|m_1\|_2^2} + \sum_{i=2}^n \left( \frac{x_i - \Phi'(s_{i-1})}{s(\sigma)} \cdot \frac{m_i^\top m_1}{\|m_1\|_2^2} \right) \right),$$

and $q'(s|t) = q(s|t)\mu^\top \Sigma^\dagger (s - \mu t)$. Equality (d) follows from Lemma 2 and independence between $X_i, X_j$ for $i \neq j$.

Let's write the SVD of $A = USV^\top$ with $U \in \mathbb{R}^{(n-1)\times(n-1)}$, $S \in \mathbb{R}^{(n-1)\times d}$, $V \in \mathbb{R}^{d\times d}$. Further, let's write $A$ in terms of the following block matrix form

$$A = [a_1 \quad A_2]$$

where $a_1 \in \mathbb{R}^{(n-1)\times 1}$ and $A_2 \in \mathbb{R}^{(n-1)\times(d-1)}$. Then,

$$\beta^2 \mu^\top \Sigma^\dagger \mu = \frac{1}{\|m_1\|_2^4} m_1^\top A^\top \left( A \left( I - \frac{m_1 m_1^\top}{\|m_1\|_2^2} \right) A^\top \right)^\dagger A m_1$$
$$= \frac{1}{\|m_1\|_2^2} a_1^\top (A_2 A_2^\top)^\dagger a_1. \tag{26}$$

Note that the vectors and matrices $a_1, A_2$ are uniquely determined by $M$ and rotational matrix $Q$. Finally, we define the function $\Psi_1(\cdot)$ (and corresponding $\Psi_i(\cdot)$) as

$$\Psi_1(M) := \frac{1}{\|m_1\|_2^2} a_1^\top (A_2 A_2^\top)^\dagger a_1.$$

Now, by virtue of the pseudo-inverse, if $A_2 A_2^\top$ is a zero-matrix, $(A_2 A_2^\top)^\dagger$ is also a zero matrix and hence $\Psi_1(M) = 0$. Otherwise, $A_2 A_2^\top$ will always have a finite nonnegative eigenvalue, and therefore $\lambda_{\max}((A_2 A_2^\top)^\dagger)$ is finite and positive. Hence, by the matrix identity $v^\top A v \leq \lambda_{\max}(A) \cdot v^\top v$ for vector $v$ and positive semidefinite matrix $A$, we can conclude that the function $\Psi_1(M)$ is always finite and non-negative. Finally, we combine with (25) to conclude that

$$\mathcal{I}_X(m_1^\top \theta = t) \leq \frac{L}{s(\sigma)} \cdot \frac{\|Mm_1\|_2^2}{\|m_1\|_2^4} + \frac{1}{\beta^2} \Psi_1(M).$$

Taking the expectation over all possible $m_1^\top \theta = t$ completes the proof. The same reasoning follows through for each $i = 2, \ldots, n$. We remark that the analysis of Fisher information $\mathcal{I}_X(m_i^\top \theta)$ is independent for different $i = 2, \ldots, n$, and so the rotations we take in the start of this section for $i = 1$ do not affect the rotations for other $i = 2, \ldots, n$.

### 4.4 Proof of Lemma 12

For the Gaussian linear model, we can have a direct analysis of Fisher information due to joint Gaussianity. In particular, we again consider $\theta \sim \mathcal{N}(0, \beta^2 \mathbf{I}_d)$. We follow the notation from before and define $A = [m_2 \quad \ldots \quad m_n]^\top$. Then, consider the joint Gaussian variable $Y := (X, A\theta, m_1^\top \theta)$. The covariance matrix $\Sigma$ of Y is given by

$$\Sigma = \mathbb{E}\left[ (Y - \mathbb{E}Y)(Y - \mathbb{E}Y)^\top \right] = \begin{bmatrix} \beta^2 AA^\top + \sigma^2 \mathbf{I}_{n-1} & \beta^2 Am_1 & \beta^2 AA^\top & \beta^2 Am_1 \\ \beta^2 m_1^\top A^\top & \beta^2 \|m_1\|_2^2 + \sigma^2 & \beta^2 m_1^\top A^\top & \beta^2 \|m_1\|_2^2 \\ \beta^2 AA^\top & \beta^2 Am_1 & \beta^2 AA^\top & \beta^2 Am_1 \\ \beta^2 m_1^\top A^\top & \beta^2 \|m_1\|_2^2 & \beta^2 m_1^\top A^\top & \beta^2 \|m_1\|_2^2 \end{bmatrix}.$$

By joint Gaussianality, from straightforward computation the conditional distribution of $X$ given $m_1^\top \theta = t$ has mean $\begin{bmatrix} Am_1/\|m_1\|_2^2 \\ 1 \end{bmatrix} t$ and covariance

$$\begin{bmatrix} \beta^2 AA^\top - \frac{\beta^2 Am_1 m_1^\top A^\top}{\|m_1\|_2^2} + \sigma^2 \, \mathrm{I}_{n-1} & 0 \\ 0 & \sigma^2 \end{bmatrix},$$

and hence, we readily calculate the Fisher information $\mathcal{I}_X(m_1^\top \theta = t)$ as

$$\mathcal{I}_X(m_1^\top \theta = t) = \begin{bmatrix} \frac{m_1^\top A^\top}{\|m_1\|_2^2} & 1 \end{bmatrix} \begin{bmatrix} \beta^2 AA^\top - \frac{\beta^2 Am_1 m_1^\top A^\top}{\|m_1\|_2^2} + \sigma^2 \, \mathrm{I}_{n-1} & 0 \\ 0 & \sigma^2 \end{bmatrix}^{-1} \begin{bmatrix} \frac{Am_1}{\|m_1\|_2^2} \\ 1 \end{bmatrix}$$

$$= \frac{1}{\sigma^2} + \frac{m_1^\top A^\top}{\|m_1\|_2^2} \left( \beta^2 AA^\top - \frac{\beta^2 Am_1 m_1^\top A^\top}{\|m_1\|_2^2} + \sigma^2 \, \mathrm{I}_{n-1} \right)^{-1} \frac{Am_1}{\|m_1\|_2^2}.$$

Then, note that

$$\sigma^2 \cdot \mathcal{I}_X(m_1^\top \theta = t) = 1 + \frac{1}{\|m_1\|_2^4} \cdot m_1^\top A^\top \left( \frac{\beta^2}{\sigma^2} A \left( \mathrm{I}_d - \frac{m_1 m_1^\top}{\|m_1\|^2} \right) A^\top + \mathrm{I}_{n-1} \right)^{-1} Am_1$$

$$\overset{(a)}{=} 1 + \frac{1}{\|m_1\|_2^4} \cdot m_1^\top A^\top \left( AW^\top W A^\top + \mathrm{I}_{n-1} \right)^{-1} Am_1$$

$$\overset{(b)}{=} 1 + \frac{1}{\|m_1\|_2^4} \cdot m_1^\top A^\top \left( \mathrm{I}_{n-1} - AW^\top (\mathrm{I}_d + WA^\top AW^\top)^{-1} WA^\top \right) Am_1$$

$$\overset{(c)}{\leq} 1 + \frac{1}{\|m_1\|_2^4} \cdot m_1^\top A^\top Am_1$$

$$= 1 + \frac{1}{\|m_1\|_2^4} \cdot m_1^\top (M^\top M - m_1 m_1^\top) m_1$$

$$= \frac{\|Mm_1\|_2^2}{\|m_1\|_2^4}.$$

In (a), the matrix $W \in \mathbb{R}^{d \times d}$ is chosen to satisfy $\frac{\beta^2}{\sigma^2} \left( \mathrm{I}_d - \frac{m_1 m_1^\top}{\|m_1\|^2} \right) = W^\top W$; from taking the SVD of $\frac{\beta^2}{\sigma^2} \left( \mathrm{I}_d - \frac{m_1 m_1^\top}{\|m_1\|^2} \right)$ one can easily find such a matrix $W$. The inequality (c) follows since

$$m_1^\top A^\top AW^\top (I + WA^\top AW^\top)^{-1} WA^\top Am_1 \geq 0.$$

Equation (b) follows from the following Woodbury matrix identity.

**Lemma 18** (Woodbury Matrix Identity). *For any two matrices $U \in \mathbb{R}^{n \times k}, V \in \mathbb{R}^{k \times n}$, $(\mathrm{I}_n + UV)^{-1} = \mathrm{I}_n - U(\mathrm{I}_k + VU)^{-1}V$.*

Finally, taking the expectation over all possible $m_1^\top \theta = t$ and dividing both sides by $\sigma^2$ concludes the proof. The same reasoning follows through for $i = 2, \ldots, n$.