[Reviews · NeurIPS 2020]

Review 1

Summary and Contributions: Minimax risk bounds for Generalized Linear Models and Gaussian Linear models for L_2 and entropic risks. The results go beyond Lee-Courtade. using tools developed by Y. Wu. Gives interesting and improved results even when design matrix is not well structured (fat, rank deficient). Works for some random designs. The paper provides evidence that information theoretic methods can extract (sense?) dependencies on the structure of design matrix.

Strengths: Well-written. Timely, interesting and explainable results. Of interest to any theoretically inclined reader. I find the connexion between risks and ratio of Schatten 1-2 norms of design matrix particularly appealing.

Weaknesses: Nothing special

Correctness: Everything looks plausible. Could not spot any error.

Clarity: Very well written. Sober and well-chosen notation. Non-ambiguous definitions. Theorems are clearly stated, Discussion is sharp and readable.

Relation to Prior Work: Yes. Section 2.1. does a very good and useful job.

Reproducibility: Yes

Additional Feedback:


Review 2

Summary and Contributions: The paper considers generalized linear models and designs newer regret bounds which offer a significant improvement in the prior art. They also consider as special case \Gaussian linear models and when the linear coefficients are sampled from a Gaussian ensemble. I am not active in this area of research, nonetheless I found the results striking and engaging.

Strengths: The analysis is impressively novel and comprehensive. The paper is primarily a theoretical exposition to improve the regret bounds, which in and of itself is a worthy contribution. The ideas have been delineated logically and clearly.

Weaknesses: In general, paper is sound. I would be interested to see what happens if we relax the Gaussian ensemble assumption of 2.2, namely, what if there is mixture of Gaussian or a heavy-tailed sampling instead of Gaussian sampling? It would be also interesting to see some intuitive justification of the losses considered in equation (5) and (6) and behind Lemma 1. Some minor points, the notation of subscripts of \theta should be introduced before stating equation (5). Post-rebuttal : I am satisfied with the responses and would like to keep the score as is.

Correctness: Seems correct.

Clarity: Very well written.

Relation to Prior Work: Satisfactory.

Reproducibility: Yes

Additional Feedback:


Review 3

Summary and Contributions: This paper is devoted to establishing tight minimax lower bounds on the prediction error in generalized linear models. Crucially, the lower bounds established in this paper require weaker spectral properties of the design matrix, i.e. robust to both near-zero or extreme values in the spectrum. The main idea behind the proof is similar to [23], where the authors first reduce to a Bayesian entropic loss, and then apply the general relationship between the mutual information and Fisher information in [24] to lower bound the Bayesian entropic loss. The main contributions of this paper mainly include: 1. A general minimax lower bound on the prediction error for a large class of generalized linear models; 2. A minimax lower bound for sparse (Gaussian) estimation which requires weaker spectral properties of the design matrix; 3. A direct Bayesian lower bound instead of going through the multiple hypothesis testing and the metric entropies.

Strengths: The assumptions in this paper are general, claims are sound, and the results are interesting. It is particularly interesting to see an explicit Bayesian lower bound based on a natural Gaussian prior, whereas most prior work only showed even a weaker bound under a carefully constructed discrete prior. Also see the listed contributions above.

Weaknesses: 1. The novelty of this paper seems questionable, mainly in view of [23]. Specifically, [23] studied a similar problem for generalized linear models where the only difference seems to be that the estimation error was considered instead of the prediction error. The technical steps are also very close to each other: both work reduced to Bayesian entropic loss, then the result of [24] was invoked to show that an upper bound on the Fisher information is sufficient, and finally the authors provided upper bounds on the Fisher information. Of course the last step is different; however this difference does not seem to add too much novelty. 2. Specializing to sparse models, the contribution that weaker spectrum properties are now sufficient seems to be outweighed. First, some problems suffered in the previous approaches can be easily fixed. For example, the authors commented that when the design matrix M has two repeated columns, the previous lower bound becomes trivial. However, this can be easily fixed as follows. Assume wlog that the first two columns of M are the same, then we may simply fix \theta_1 = 0 and allow others to vary arbitrarily. In this way we effectively remove the first column of M, keep the same sparsity property, and only reduce the parameter dimension from d to d-1. Now applying the previous lower bound to the new and well-conditioned matrix gives the desired minimax rate. Also note that similar approaches can be taken even when half of the columns of M are repeated (and we reduce the dimension from d to d/2, which does not affect the rate analysis). Therefore, this comparison may seem slightly unfair and does not make a too strong case to me. Second, I do not fully understand why the authors treat the assumption k < n in previous work a ``crucial disadvantage". Note that for a typical constant noise level \sigma, the previous lower bound already shows that a sample complexity of n > k*log(ed/k) is necessary to achieve a constant statistical error. Also, for any \sigma, Eqn. (12) in this paper also shows that the case n > k gives a trivial error \sigma^2. This seems to suggest that one may wlog restrict to n > k for showing minimax lower bounds. 3. The tightness of the result is not sufficiently discussed. The authors claimed that their Theorem 3 is tight if either the largest or the smallest eigenvalues are of the same order. However, it seems that in those scenarios the previous lower bounds also give the tight answer. In other words, the authors did not explicitly construct an example such that the previous lower bound is not tight but the current bound becomes tight. Moreover, compared with the lower bound in (10), the authors did not show that the new bound provides a uniformly improvement over it. Finally, and most importantly, there is a missing logarithmic factor in the current lower bound, which is known to be necessary and important in sparse estimation. So missing the log factor seems to make the bound not very desirable in my opinion. Post rebuttal: The points #2, #3 are satisfactorily addressed in the rebuttal; please add these discussions to the final paper. However, my novelty concern over [23] is not adequately addressed. I took a closer look at both papers, and the only difference is on the upper bounds of the Fisher information, while other steps (bayes entropic loss, generalized van-trees inequality) are essentially the same. I agree that the current paper uses a different approach to upper bound the Fisher information: [23] used a Jensen's inequality (or a data-processing property of Fisher information) to relate the individual Fisher information to the trace of the entire Fisher information matrix (p.s. I do not understand why the authors call this a "single-letterization"); in the current paper, the individual Fisher information is studied directly by assuming a Gaussian prior and using the rotational invariance of the Gaussian distribution. However, I would prefer to treat this step as a direct and relatively straightforward computation of the Fisher information, and still do not think there is much technical innovation here. Given that this novelty concern remains, I decide to only increase my score from 5 to 6.

Correctness: The main proofs look correct to me.

Clarity: Yes.

Relation to Prior Work: The related literature was sufficiently discussed, but the authors should make more clarifications on the added novelty compared to [23].

Reproducibility: Yes

Additional Feedback: Some minor comments: Eqn. (11): the first \sup should be \inf. Sharpness: it might be better to clarify that the tight answer when the largest eigenvalues are of the same order is given by the first bound, and that when the smallest eigenvalues are of the same order is given by the second bound.


Review 4

Summary and Contributions: The authors establish a lower bound for the minimax prediction loss of generalized linear model through entropic loss.

Strengths: It improves on the current literature when the design matrix is ill-posed. They then discuss the situation for the Gaussian linear model and Gaussian design matrix.

Weaknesses: The contribution of the paper compared to its closely related work [19] seems to incremental. -- Update: the author's explanation partially addressed my concern.

Correctness: Probably. Did not check all details.

Clarity: Yes.

Relation to Prior Work: Based on my own reading, also as explained in the paper, the paper is very closed to reference [19]. The only improvement is that [19] considers estimating of entire $M\theta$, and here they consider \theta first and then estimate M\theta. The contribution of the paper seems to be limited.

Reproducibility: Yes

Additional Feedback:

[Author Response · NeurIPS 2020]

We thank all reviewers for their valuable comments. We thank R1 for positive comments on the writing style, R2 for
recognizing the novelty of our work, R3 for raising insightful questions and concerns, and R4 for encouraging us to
clarify differences from previous literature.

**1. (R1, R2, R3, R4) Uniform improvement**. We point out that, up to the logarithmic factor, **our bound uniformly im-**
**proves upon previous results of the form (10)**. To see this, let $M'$ be the realization of $\Phi_{2k,+}(M)$ with most columns
($M'$ will have at least $k$ columns since adding a column to $M'$ will not decrease its maximum singular value). Note
that $k\Phi_{2k,-}(M)/\Phi_{2k,+}(M) \leq k\sigma_{\min}^2(M')/\sigma_{\max}^2(M')$ by definition of $\Phi$. We may assume that $M'$ has full column
rank, otherwise the proof is trivial. Then, for any submatrix $Q \in \mathbb{R}^{n \times k}$ of $M'$, we have $k\sigma_{\min}^2(M')/\sigma_{\max}^2(M') \leq$
$k\sigma_{\min}^2(Q)/\sigma_{\max}^2(Q)$ (see, e.g., 2.2.33. of [Hanson–Lawson, "Extensions and Applications of the Householder Al-
gorithm for Solving Linear Least Squares Problems", 1969]). Next, $k\sigma_{\min}^2(Q)/\sigma_{\max}^2(Q) \leq \|\Lambda_Q\|_1^2/\|\Lambda_Q\|_2^2$ holds
because $\sigma_{\max}^2(Q)\|\Lambda_Q\|_1^2 \geq k\sigma_{\min}^2(Q)(\sigma_{\max}^2(Q)\|\Lambda_Q\|_1) \geq k\sigma_{\min}^2\|\Lambda_Q\|_2^2$. Combining these with Corollary 4 gives the
claim. Our result, although without a logarithmic term, elegantly captures the spectral dependence and can provide
much sharper bounds when the minimum and maximum constrained eigenvalues differ in order. Our results also give
interesting bounds when the minimum constrained eigenvalue is degenerate (i.e., equal to 0).

**2. (R3) Tightness of result**. Previous result (10) is tight *only* when *all* the eigenvalues of the spectrum are of the same
order, while our bound is tight when the top $\Theta(d)$ eigenvalues are of the same order *or* when the lower $\Theta(d)$ eigenvalues
are of the same order. An example is when the minimum constrained eigenvalue of $M$ is close to zero (say, $1/t$ for
some large $t$), while the maximum constrained eigenvalue is equal to $t$. Then, (10) would scale as $d/nt^2$, ignoring log
terms. With our result, if the largest $d/2$ eigenvalues are of the same order (say, proportional to $t$), our result scales as
$d/n$, *regardless of how small the remaining eigenvalues are*. On the other hand, if the smallest $d/2$ eigenvalues are
of the same order (say, proportional to $1/t$), our result also scales as $d/n$, *regardless of the remaining eigenvalues*. In
general, our result tolerates extremal eigenvalues in the spectrum, as shown in Table 1. **Log factor**. We would argue
that the improved spectral dependence takes precedence over the log factor, which is insignificant in many cases. For
example, when $k$ is not very sparse, say $k = \Theta(d^c)$ for some $c \in (0, 1]$, the log factor is not significant and our results
can be orders better than (10) when matrix $M$ is mildly ill-conditioned. In the very sparse case, say $k = O(\log d)$, our
results still provide meaningful bounds for $M$ with minimum constrained eigenvalue close to 0.

**3. (R3) Dimensionality reduction**. For the sparse model with $M$ having repeated columns, Reviewer 3's point on
reducing the dimensionality via ignoring certain components of $\theta$ is valid. However, the result of (10) still depends on
the ratio between minimum and maximum constrained eigenvalues of the new "effective" matrix, and leads to a poor
lower bound when the minimum and maximum constrained eigenvalues are of a different order. Another point is that
for a general matrix $M$ with zero singular values, it is not immediately clear if the same dimensionality reduction trick
can be used. This is because of the sparsity constraint on $\theta$ (namely, $\|\theta\|_0 \leq k$); for example, an approach one may try
is to use an orthogonal matrix $U$ and map $\theta \mapsto U\theta$ and $M \mapsto MU^\top$ so that the matrix $MU^\top$ has repeated columns or
zero columns, and proceed to ignore certain components of $U\theta$. Yet, in general restricting $U\theta$ to satisfy $\|U\theta\|_0 \leq k$
does not necessarily imply $\|\theta\|_0 \leq k$, and hence, we cannot map the problem to one where we can easily use the same
dimensionality reduction method on $(M, \theta) \mapsto (MU^\top, U\theta)$. Moreover, in general when the spectrum of $M$ is all
positive (with divergent large/small eigenvalues), one cannot use dimensionality reduction to improve the result of (10).

**4. (R3) Case $n < k$**. It should be noted that the result of (10) depends on the ratio of restricted eigenvalues and hence
cannot be used to give a meaningful lower bound when $n < k$ due to a degenerate minimum restricted eigenvalue of 0.
From this perspective, it is still interesting to have reasonable lower bounds for this situation. Our results accomplish
this—despite the trivialness of the error, our method successfully provides a tight lower bound for the Gaussian design.

**5. (R3) Novelty with respect to [23]**. We consider the prediction scheme as opposed to the estimation scheme
considered in [23]. In terms of technical novelty, the key step in [23] requires a single-letterization bound of the Fisher
information (i.e., Lemma 9 of [23]). The prediction problem is fundamentally different (and often most important in
practice) and the same single-letterization cannot be applied. In our setting, we developed new bounds for the expected
Fisher information (i.e., Lemmas 10 & 11), with proofs essentially orthogonal to [23]; see supplementary for details.

**6. (R4) Improvement with respect to [19]**. We would like to note that the results of [19], like (10), depend on the
ratio between smallest and largest constrained eigenvalues (defined as "$\tau[\cdot]$" in Theorem 2 of [19]). Hence, the results
of [19] suffer from the same (undesirable) structural dependencies as (10). Our bounds overcome this dependence, and
we have showed evidence that our bound provides nontrivial improvements in the above points 1. to 4. and Section 2.1.

**7. (R2) Other extensions and intuition**. In general, one may derive minimax bounds for other settings such as
subexponential designs and heavy-tailed designs with spectral concentration results. The gap of Lemma 1 comes from
$L_2$ bounds arising from bounding entropy with the entropy of a Gaussian having same variance; in fact, our entropic
minimax bounds can be extended to other norms (not only $L_2$) via tools from rate distortion theory (see, e.g., [17]). We
will add remarks on this point in any revision.

[Meta-Review · NeurIPS 2020]

The reviewers enjoyed reading your paper, and have made several useful comments for the camera-ready version. It would be particularly useful for readers if the authors were clear on the precise technical differences between their work and the work of Lee-Courtade, which they follow closely.